# A recurring disease outbreak following litchi fruit consumption among children in Muzaffarpur, Bihar—A comprehensive investigation on factors of toxicity

**Sukesh Narayan Sinha**[1]***, Ungarala Venkat Ramakrishna**[1]**, P. K. Sinha**[2]**, C. P. Thakur**[3]

**1** Food Safety Division, National Institute of Nutrition, Hyderabad, Telangana, India, **2** RMRI, Patna, India, **3** Balaji Utthan Sansthan, Patna, Bihar, India

* sukeshnr_sinha@yahoo.com

**Data Availability Statement:** All relevant data are within the manuscript.

## Abstract

Litchi fruits are a nutritious and commercial crop in the Indian state of Bihar. Litchi fruit contains a toxin, methylene cyclopropyl-glycine (MCPG), which is known to be fatal by causing encephalitis-related deaths. This is especially harmful when consumed by malnourished children. The first case of litchi toxicity was reported in Bihar in 2011. A similar event was recorded in 2014 among children admitted to the Muzaffarpur government hospital, Bihar. Litchi samples sent to ICMR-NIN were analyzed and MCPG was found to be present in both the pulp and seed of the fruit. Diethyl phosphate (DEP) metabolites were found in the urine samples of children who had consumed litchi fruit from this area indicating exposure to pesticide. The presence of both MCPG in litchi and DEP metabolites in urine samples highlights the need to conduct a comprehensive investigation that examines all factors of toxicity.

## 1. Introduction

Litchi (*Litchi chinensis Sonn.*) is a tropical and subtropical fruit belonging to the genus Litchi in the soapberry family, Sapindaceae. Mainly produced in China, India, and Vietnam, litchi fruits are highly perishable and have a very short harvest period (approximately 1 month) [1]. The crop is highly susceptible to many diseases, which can lead to significant losses in quality and yield [2]. To overcome this limitation, pesticides are commonly used.

In India, litchi fruit is one of the most prominent agricultural products in the northern state of Bihar [3]. In 2012–2013, a total of 7 metric tons/hectare (MT/ha) were produced in India, and Bihar was the leading producer, with 44.2% of the yield (Indian Horticulture Database 2013, National Horticulture Board). During the same period, this region was also responsible for using a significant portion of the total pesticide imported and produced in India [4]. Since pesticides are known to be carried from crops into our diets, they raise a significant concern toward human health.

The first outbreak of litchi toxicity occurred in 2011, in which 14 children died following litchi consumption in Bihar. Investigation of the deaths revealed that the cause was similar to

**Funding:** No specific funding has been received for this study.

**Competing interests:** The authors have declared that no competing interests exist.

pesticide toxicity [5]. Later, a similar case occurred in Kolkata; however, the cause of death was identified as a toxin, methylene cyclopropyl-glycine (MCPG). MCPG is found in unripe litchi fruit and depletes glucose reserves in the body, making it more fatal for undernourished children, triggering encephalitis-related deaths. These findings led to the recommendation that children should not consume unripe litchi fruit [6].

Recent evidence has shown MCPG is found in the seeds of litchi fruit, and in the presence of platinum oxide ($PtO_2$), MCPG hydrogenates into a blend of leucine, isoleucine, norleucine, and two isomeric $\alpha$ -(methyl cyclopropyl) glycines [7]. The IR and NMR spectra of the newly found compound showed a doled-out structure and the $\alpha$-MCPG demonstrated hypoglycemic activity when infused dermally into mice. No accurate correlation of the hypoglycemic effects of the new acid and its advanced homolog, hypoglycin A, was made [8].

The first acute encephalitis syndrome (AES) outbreak occurred in Eastern India in 1973 [8, 9]. However, the AES outbreak in 2011 was centered in North Bihar, with several cases identified in each of the affected towns. The average age of the patients was 5 years (3 months–10 years) and disease onset correlated with the natural production season of litchi. The first case was reported at the Krishnadevi Deviprasad Kejriwal Maternity Hospital (KDKM) on June 11, 2011. There was a record of 26 deaths, representing 31% of the total number of children admitted [10]. The case casualty rate was 20% in Gorakhpur, India and 25% from the 2004–2009 pandemic in Vietnam [11] and North India [12]. and most belonged to the lower socioeconomic sections of society. In the epidemic at Gorakhpur, India, 93.69% of cases were among children aged 1–5 years [12]. The proportion of males to females was reported as 1.45:1 in Gorakhpur [13] and 1.2:1 in Vietnam [11].

The death of the children led to the focus on the toxicity of MCPG present in the fruit, but the role of pesticides has not been examined. More investigation is needed to identify the role of pesticides in the death of the patients. Those farming the crop stated that pesticides containing cypermethrin were used on the litchi trees to prevent bats and insects damaging the fruit. The excessive use of these agrochemicals and harvesting the fruits before the expiry of the sprayed pesticides could also have a more considerable impact on the children who ingested them [13].

The National Vector Borne Disease Control Program (NVBRC Program), India has reported that concentrations of $\alpha$-cypermethrin, a largely active pyrethroid insecticide, above the maximum limit for humans were found in the litchi samples collected in the affected areas. It was alleged that there could be a link between the AES outbreak and pesticide poisoning [14].

Orchard leaseholders also revealed that they employed residents from the community as caretakers to work in the litchi farms. Among the 14 case patients reported, 3 were involved in the application of pesticides and 8 lived together with individuals who were employed in the orchards during the outbreak period [15].

Different types of pesticides have been reported to have been used on litchi trees. Chlorpyrifos, imidacloprid, and deltamethrin were used for controlling seed-borne diseases while other pesticides such as carbaryl, dolofase, phosphamide, chlorpyrifos, and dichlorobis were also used for cultivation, as reported by the National Research Centre for Litchi, Muzaffarpur [16]. Hence, the presence of pesticides should also be investigated to obtain clarity regarding the causes of toxicity that led to death in the children in Muzaffarpur.

In 2014, more cases of children suffering from the effects of litchi fruit toxicity were identified in Bihar. The local government hospital reported children of different age groups being admitted. To comprehensively understand the reasons underlying the toxicity, nine urine samples and the litchi fruit samples were investigated in our laboratory. The present investigation

aimed to determine the toxin compounds present in the fruit samples and the urine samples of the victims.

## 2. Materials and methods

### 2.1 Chemicals and reagents

The following materials were purchased from Sigma Aldrich (USA): acetonitrile, methanol, 1-butanol, LC-MS grade, ninhydrin reagent 2%, glacial acetic acid, and TLC plates (aluminum sheets). Formic acid (analytical grade) was obtained from Merck, USA. To detect pesticide residues in food, QuEChERS (a mixture containing magnesium sulfate, sodium acetate along with primary secondary amine, PSA, tubes) was purchased from Agilent (USA), as was the liquid chromatography (LC) column. Standards of pesticides, insecticides, and herbicides were purchased from Sigma Aldrich. All reagents and standards were prepared fresh in LC-MS grade water or solvent before use. The standards of organophosphate pesticide (OP) metabolites, diethyl dithiophosphate, diethyl thiophosphate, dimethyl thiophosphate, dimethyl dithiophosphate, dimethyl phosphate, and diethyl phosphate (DEP) were obtained from the National Centre for Environmental Health, CDC (Atlanta, USA).

### 2.2 Collection of litchi samples

Litchi fruit samples were collected from Balaji Utthan Sansthan, Patna, Bihar, the same area as that of the affected children. Samples were subsequently sent to the Indian Council of Medical Research-National Institute of Nutrition (ICMR-NIN) Laboratory, Hyderabad, Telangana for analysis.

### 2.3 Collection of urine samples

Urine samples (n = 9) were collected from children admitted with suspected litchi toxicity to Shri Krishna Medical College and Hospital, Muzaffarpur Balaji University, and sent to the ICMR-NIN laboratory for further analysis. Written informed consent was obtained from parents of all children, and experiments were conducted under the institutional ethical guidelines of Balaji Utthan Sansthan.

All experimental protocols and procedures were approved (approval number: 2014/IED/Proj/001) by the institutional committee of Balaji Utthan Sansthan.

### 2.4 Analysis of OP metabolites in urine samples

Analysis of urine samples for the relevant metabolites can provide critical information for the estimation of human exposure to OP pesticides. Urine samples were used for the analysis of OP pesticide exposure in the affected children, with samples analyzed for nonspecific dialkyl phosphate metabolites. For the estimation of OP metabolites, urine was selected because collection is simple and non-intrusive, metabolite concentrations are normally significantly higher than those in blood, and laboratory methods are accessible to quantify these metabolites at low detection levels. Urine samples were lyophilized according to the method described by Sinha et al. (2014). The analysis was conducted using liquid chromatography mass spectrometry (LC-MS/MS), and the identification of pesticide metabolites in urine was performed using a previously developed described by Sinha et al. (2014) [17].

### 2.5 Instrumentation

A 4000-QTRAP triple quadrupole hybrid mass spectrometer (Applied Biosystems MDS Sciex, USA) aligned with ultra-fast liquid chromatography with LC 20 AD, binary pump (ultra-fast

liquid chromatography (UFLC), Shimadzu) was used for the identification and estimation of OP pesticide metabolites (six OP metabolites). The analysis was performed in negative turbo ion spray (ESI) mode. The reverse-phase C-18 column was used for LC.

## 2.6 Liquid chromatography–mass spectroscopy conditions (LC-MS)

An UFLC instrument outfitted with a triple-quadrupole ion trap mass spectrometer in operation inside the multiple reaction monitoring (MRM) mode was utilized for this investigation. Chromatographic separation was accomplished for the six OP metabolites by using a liquid chromatograph equipped with an SB-C18 reverse phase column with a particle size of 1.8 μm and dimensions of 150 × 4.6 mm. Extracted urine samples (10 μL) were administered into the instrument utilizing a Shimadzu auto-sampler fitted with a Hamilton 100 μL syringe. Gradient compositions of 0.1% acid in water (gradient-A) and acetonitrile (gradient B) at a flow rate of 0.2 mL were utilized. nonetheless, the best affectability and separation for all the pesticides were accomplished utilizing the gradient compositions [17].

The analytes were investigated in the MRM negative ESI mode with high resolution. The interface radiator was maintained at a temperature of 550˚C and an ion-spray (IS) voltage of 5500 eV. A full autotune of the mass spectrometer was performed prior to each sample set investigation. A full scan of the mass spectra of all pesticides was recorded to select the optimal mass to charge quantitative relation (m/z) molecule (Q1) victimization continuous infusion of each pesticide inside the positive ionization method of ESI. The product mass spectra were acquired with the nonstop infusion of each analyte; therefore, Q1, which was compared to the protonated precursor molecule, remained constant. The most abundant product ion for each compound was then assigned for MRM examination. At least two of the most intense product ions were separated; one ion was utilized for quantification, while the other was utilized for affirmation, according to the three head rules given for mass spectrographic investigations of OP pesticides [18, 19]. The optimization of the source-dependent parameters, such as nebulizing gas (GS1), heating gas (GS2), and curtain gas, were analyzed in the flow injection analysis mode. The GS1, GS2, and curtain gas pressures were maintained at 35, 40, and 25 psi, respectively, throughout the study. In addition, the de-clustering potential (DP), collision energy (CE), entrance potential (EP), and collision exit potential (CXP) were utilized according to the predefined affectability of the strategy [17].

## 2.7 Extraction of urine samples

Using a lyophilizer, 10 mL of each urine sample was transferred into 20 mL test tubes and freeze-dried. To each lyophilized test tube, 5 mL of acetonitrile was added and vortexed for 10 min and then moved into 15 mL centrifuge tubes (first concentrate). The samples were again extracted using 5 mL of acetonitrile for 2 min and then moved to the acetonitrile-extracted sample arranged in the first step. When the extraction was completed, the contents of the tubes were centrifuged for 8 min at 3000 rpm. In this procedure, the supernatant was collected into another 15 mL tubes for drying and further positioned in a TurboVap at 20˚C under 15 psi of nitrogen to ensure that it was completely dried. The tubes were reconstituted using 1 mL acetonitrile for analysis [17].

## 2.8 Mass spectrometer instrument control by the analyst program

The ABSciex Analyst 4.1.2 software package controlled the MRM operational mode for the 4000 QTRAP mass spectrometers. The Analyst software system additionally controlled the operational procedure of the instrument. This program set the instrument to acquisition by ESI in the negative mode for the analysis of the varied analytes. Different DP, CE, EP, and

CXP parameters were applied for the choice of confirmation and quantification of ions of every analyte depending on their physical and chemical properties. Precursor ions were isolated using the most sensitive, specific, and linear dynamic range. The product ions of the precursor ions were then used for confirmation [17].

## 2.9 Extraction of MCPG

Each part of the fruit was crushed separately in a mortar and pestle. The crushed parts were macerated with 95% (v/v) ethanol. The whole system was then allowed to stand for 3–5 d at room temperature (37°C). The slurry was pressed to collect the extract, which was filtered and evaporated under vacuum. The concentrate was diluted four times with water. A bulky precipitate formed immediately and precipitation was completed by storing the diluted extract overnight at 4°C. This was then removed by continuous rotator centrifugation and discarded.

Amino acids extracted from the samples were used for spotting on the TLC plates. The samples were spotted and run on a mobile phase in the TLC chamber. The mobile phase used for the process was 1-butanol, glacial acetic acid, and water in a ratio of 4:1:1. The system was run until the mobile phase reached three-fourth of the plate. The TLC plate was then air-dried entirely and placed in an oven for 30 min at 60°C. The dried TLC plates were sprayed with a 2% ninhydrin solution and dried again in an oven at 60°C for 45 min. Purple spots on the plate indicated the presence of glycine-like amino acids.

## 2.10 Thin Layer Chromatography (TLC) analysis

TLC aluminum sheets of 20 × 20 dimension were utilized for the identification. These plates were kept in an oven at a temperature of 120°C for 30 min to activate the silica on the sheets. The TLC chamber was saturated with the mobile phase at a ratio of 4:1:1 of 1-butanol, glacial acetic acid, and water. The extracted samples of different parts of the fruit were spotted using a capillary tube toward the bottom of the plate at 2 cm above the surface. The spots were left to dry at room temperature for 2 min. The plates were then placed inside the TLC chamber for separation and identification. The mobile phase was allowed to increase to two-third of the height of the plate and then removed from the chamber to dry at room temperature. After drying, the plates were sprayed with 2% ninhydrin solution and dried in an oven at 60°C for 45 min to observe any colored spots for identification.

## 3. Results

Each fruit was separated into three parts: fruit, seed, and rind. Fruit parts were analyzed for traces of MCPG and metabolites of different pesticides. Urine samples of the affected children were also analyzed for the same pesticides.

MCPG has been newly characterized as a component of a higher plant being isolated from seeds of Litchi chinensis. MCPG was converted by catalytic hydrogenation in the presence of platinum oxide into a mixture of leucine, isoleucine, norleucine and two isomeric α-methyl cyclopropyl glycines. All the parts of the fruit were subjected to TLC as a preliminary test to check for glycine like amino acid complexes.

### 3.1 Isolation of MCPG

The isolation of MCPG was performed using a TLC system. The solvent was run to 15 cm on the TLC plate and dark spots were observed for seed, pulp, and peel at 7.5 cm and 10.5 cm and the Rf values calculated for the analysis were found to be 0.5 and 0.7 (Fig 1). A light spot at a distance of 4 cm was observed for the seed sample with an Rf value of 0.26, which is similar to

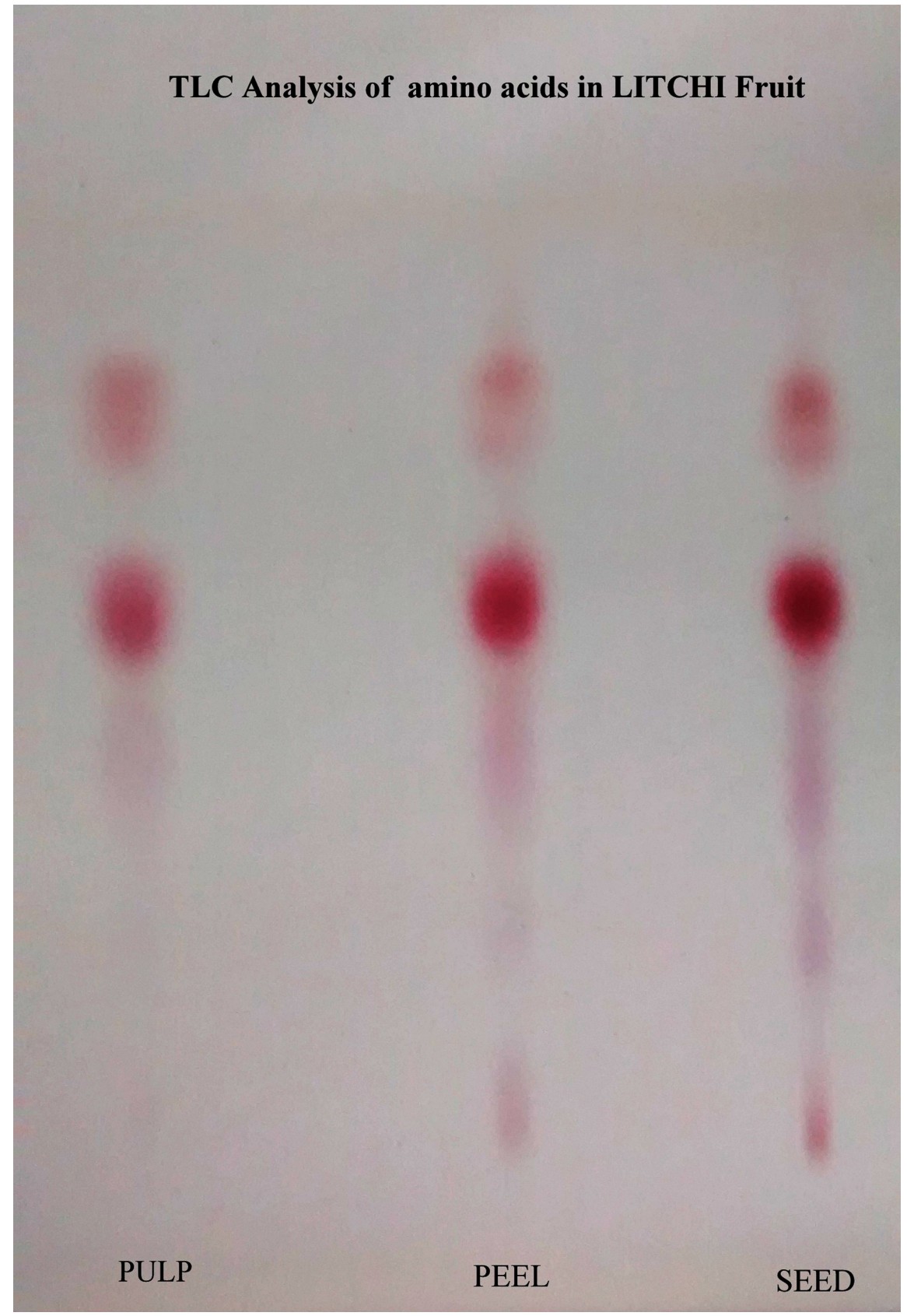

**Fig 1. TLC identification of amino acids in different parts of litchi fruit.**

the Rf value of glycine, isoleucine, and lysine-like compounds, indicating the presence of MCPG in the seed of litchi fruit.

### 3.2 MS/MS analysis

The litchi fruit was preliminarily tested for MCPG and then subjected to MS/MS to confirm its presence. The conditions used in the analysis of the sample were: DP—85, CE—22, CXP—10, Ion Source Gas—25, IS voltage - 5500v, Curtain Gas—10, Entrance potential—10. MS/MS of the samples showed chromatographic peaks at 127.2 and 127.0 for seed and pulp, respectively. The mass spectra of MCPG were found to be 127.4, indicating the presence of MCPG in our sample. The peaks are shown in Fig 2A and 2B, respectively.

### 3.3 OP pesticide metabolite analysis in urine samples

Pesticide metabolites in urine were identified using the method previously developed by Sinha et al. (2014). Urine samples (n = 9) were analyzed for the detection of DEP metabolites. The samples were extracted using acetonitrile solvent and the estimation was carried out using LC-MS. Out of the nine samples analyzed, only one was detected with high concentrations of DEP (3327 ng/mL), whose retention time was 16.5 min and the chromatographic peak (MRM) is shown in Fig 3. Further, the precursor and product ion of the identified DEP metabolite was isolated at m/z 152.9 (Q1) and 78.9 (Q2), and the conformation ion was identified at m/z 125.

## 4. Discussion

In 2014, the incidence of litchi consumption-related toxicity at Muzaffarpur, Bihar, among children led to the present investigation. Although similar studies were previously conducted, they focused on identifying MCPG, a known toxin found in litchi. However, the present study considered other sources of toxicity that can occur due to litchi consumption. The use of OP pesticides in Bihar for litchi cultivation called for a thorough investigation. Therefore, litchi fruit samples and urine samples from the children from Muzaffarpur were collected and analyzed in the present study to critically investigate and identify the toxic compounds (MCPG and OP pesticide metabolites), which may have caused the fatalities.

The preliminary analysis of all the parts of the litchi fruit, including pulp, peel, and seed, using the TLC method, showed the presence of glycine, indicating that the fruit samples contained MCPG. Furthermore, to corroborate this finding, MS/MS analysis was carried out, which indicated the presence of MCPG in both the pulp and seed of the fruit. MCPG is known to interfere with biological processes such as gluconeogenesis and fatty acid β-oxidation. Therefore, when a malnourished child with reduced hepatic stores (which is common due to low food consumption) consumes litchi fruit, hypoglycemia occurs rapidly due to the presence of MCPG. Additionally, MCPG blocks gluconeogenesis and β-oxidation of fatty acids, which further results in severe refractory hypoglycemia. MCPG is also known to cause accumulating aminoacidemia, which may lead to encephalopathy [20].

The outbreak of such an epidemic was previously linked to the consumption of litchi fruit [21–23]. These studies considered the responsible toxic chemical in the litchi fruit as MCPG and associated consuming the fruit with hypoglycemia and encephalopathy. Further, studies have determined the safe tolerable dosage of litchi fruit by considering the cumulative minimum concentration MCPG and its $LD_{50}$ values in the rat. Thus, based on the equivalent dose-

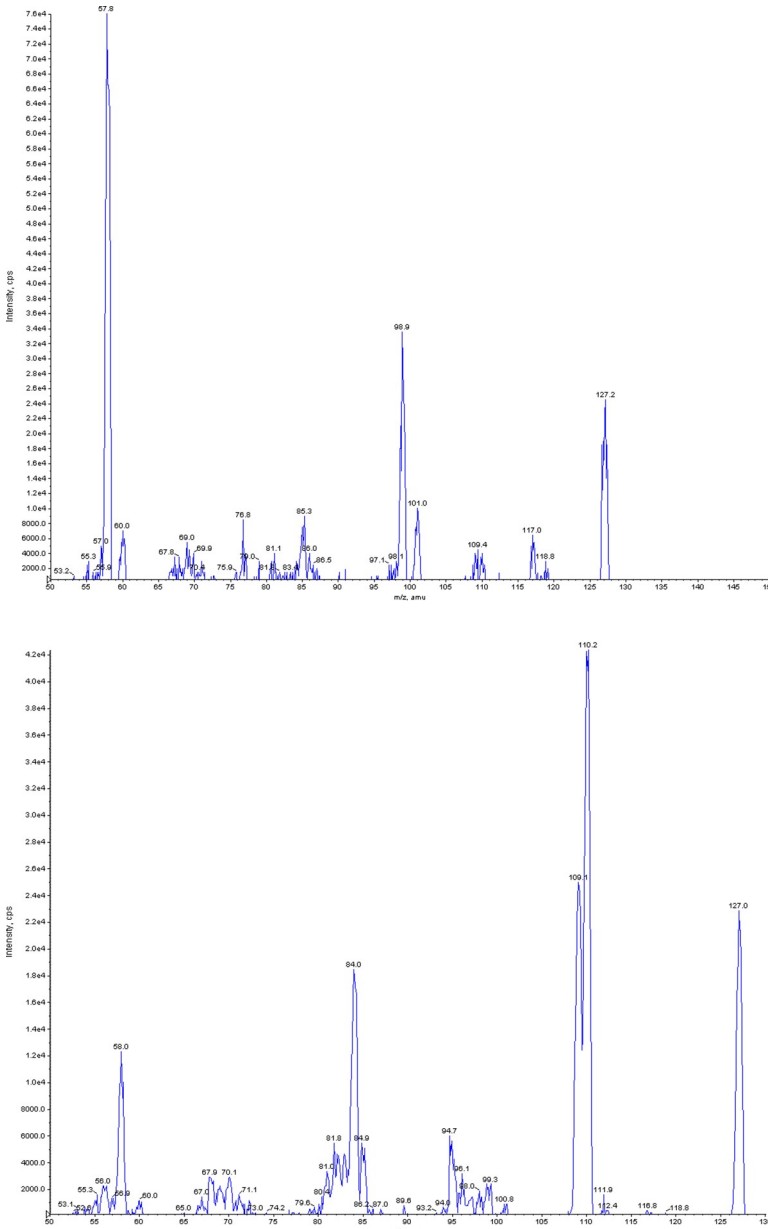

**Fig 2.** a. Mass spectra of MCPG in litchi seed. b. Mass spectra of MCPG in litchi pulp.

quantity conversion, approximately 3.9 kg of litchi pulp/day for an adult human (weighing 60 kg) and 0.59–1.17 kg of pulp/day for children (1–5 years of age, respectively) could be considered a safe quantity [24].

To obtain a complete picture of toxicity in the children, we also studied the presence of any OP metabolite in the urine samples. Usage of OP pesticides on litchi fruits during cultivation is a common practice. The analysis of the samples revealed the presence of DEP metabolites in the urine indicating pesticide exposure in the children. Thereby, the effects of OP pesticides along with those of MCPG should be considered when assessing litchi consumption-related toxicity.

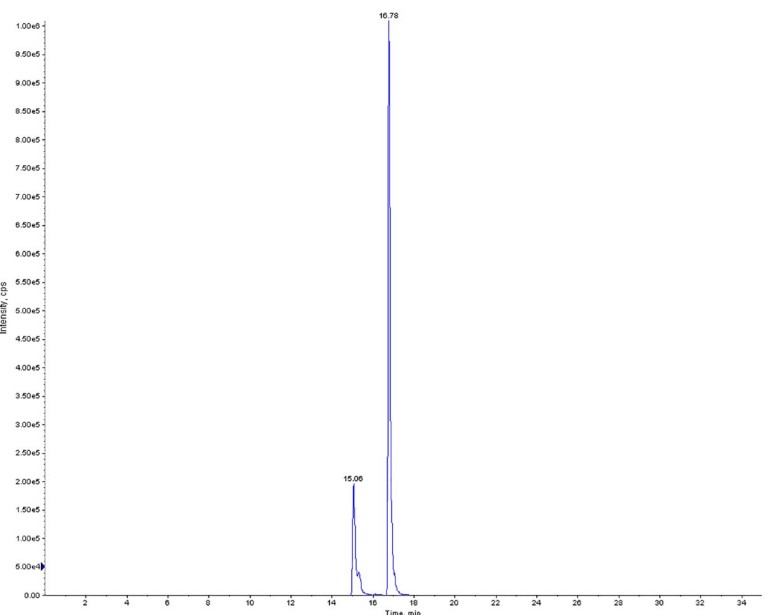

**Fig 3. Chromatograph showing a high concentration of DEP in the urine sample.**

Previously, investigations have been conducted in areas of Bihar, where the outbreak due to litchi consumption was reported. Controversial conclusions have emerged regarding the causative mechanism of death due to litchi consumption. Nevertheless, MCPG is a known chemical present in litchi, which can be fatal when consumed in excess quantities by undernourished individuals. In addition, the present study considered the other probable toxic compounds in litchi, that is OP pesticides. It can be concluded that this seasonal outbreak can be attributed to both MCPG and OP pesticides used in the cultivation of litchi crops. Useful data can be obtained if further toxicity assessments are designed in line with the present study to consider all the possible toxin compounds in areas where litchi production is predominant.

## Acknowledgments

The authors would like to take this opportunity to express heartfelt gratitude to the Director General, Indian Council of Medical Research (ICMR) and Director, ICMR—National Institute of Nutrition for the support and for providing facilities to work on the project. The author would also like to specially thank K. Vasudev & Dr. G. Subba Rao, Scientist-F, NIN for his constant support.

## Author Contributions

**Conceptualization:** Sukesh Narayan Sinha, P. K. Sinha, C. P. Thakur.

**Data curation:** Sukesh Narayan Sinha, Ungarala Venkat Ramakrishna.

**Formal analysis:** Sukesh Narayan Sinha, Ungarala Venkat Ramakrishna.

**Funding acquisition:** Sukesh Narayan Sinha, C. P. Thakur.

**Investigation:** Sukesh Narayan Sinha, C. P. Thakur.

**Methodology:** Sukesh Narayan Sinha, Ungarala Venkat Ramakrishna, P. K. Sinha.

**Project administration:** Sukesh Narayan Sinha, P. K. Sinha, C. P. Thakur.

**Resources:** Sukesh Narayan Sinha.

**Supervision:** Sukesh Narayan Sinha, P. K. Sinha, C. P. Thakur.

**Validation:** Sukesh Narayan Sinha.

**Visualization:** P. K. Sinha, C. P. Thakur.

**Writing – original draft:** Ungarala Venkat Ramakrishna.

**Writing – review & editing:** Sukesh Narayan Sinha, Ungarala Venkat Ramakrishna, P. K. Sinha, C. P. Thakur.

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
