## [Decision Letter · Decision Letter 0]

28 Sep 2020

PONE-D-20-25356

A recurring outbreak of litchi fruit consumption among children of Muzaffarpur, Bihar

PLOS ONE

Dear Dr. Sinha,

Thank you for submitting your manuscript to PLOS ONE. After careful consideration, we feel that it has merit but does not fully meet PLOS ONE’s publication criteria as it currently stands. Therefore, we invite you to submit a revised version of the manuscript that addresses the points raised during the review process.

We look forward to receiving your revised manuscript.

Kind regards,

Ratnasekhar Ch, Ph.D.

Academic Editor

PLOS ONE

Journal Requirements:

"Informed consent was obtained for all the children from their parents and all experiments were conducted under the institutional ethical guidelines by Balaji Utthan Sansthan, Patna, Bihar, India.

We confirm that all the experimental protocols and procedures were approved by a Balaji Utthan Sansthan institutional committee, Balaji Utthan Sansthan, Patna, Bihar, India. ". 

i) Please provide additional details regarding participant consent. In the ethics statement in the Methods and online submission information, please ensure that you have specified what type you obtained (for instance, written or verbal, and if verbal, how it was documented and witnessed). If your study included minors, state whether you obtained consent from parents or guardians. If the need for consent was waived by the ethics committee, please include this information.

ii) Once you have amended this/these statement(s) in the Methods section of the manuscript, please add the same text to the “Ethics Statement” field of the submission form (via “Edit Submission”).

3.  We note that Figure [1] in your submission contain [map/satellite] images which may be copyrighted. All PLOS content is published under the Creative Commons Attribution License (CC BY 4.0), which means that the manuscript, images, and Supporting Information files will be freely available online, and any third party is permitted to access, download, copy, distribute, and use these materials in any way, even commercially, with proper attribution. For these reasons, we cannot publish previously copyrighted maps or satellite images created using proprietary data, such as Google software (Google Maps, Street View, and Earth). For more information, see our copyright guidelines: http://journals.plos.org/plosone/s/licenses-and-copyright.

1.     You may seek permission from the original copyright holder of Figure [1] to publish the content specifically under the CC BY 4.0 license.  

4. Please upload a new copy of Figures 2 and 3 as the detail is not clear. Please follow the link for more information: https://blogs.plos.org/plos/2019/06/looking-good-tips-for-creating-your-plos-figures-graphics/" https://blogs.plos.org/plos/2019/06/looking-good-tips-for-creating-your-plos-figures-graphics/

Reviewers' comments:

Reviewer's Responses to Questions

**Comments to the Author**

1. Is the manuscript technically sound, and do the data support the conclusions?

Reviewer #1: Partly

Reviewer #2: Yes

Reviewer #3: Yes

2. Has the statistical analysis been performed appropriately and rigorously? 

Reviewer #1: No

Reviewer #2: Yes

Reviewer #3: Yes

3. Have the authors made all data underlying the findings in their manuscript fully available?

Reviewer #1: Yes

Reviewer #2: Yes

Reviewer #3: Yes

4. Is the manuscript presented in an intelligible fashion and written in standard English?

Reviewer #1: No

Reviewer #2: Yes

Reviewer #3: Yes

5. Review Comments to the Author

Reviewer #1: The work in the manuscript entitled, “recurring outbreak of Litchi fruit consumption among children of Muzaffarpur, Bihar” is very interesting and have high significance. However, the manuscript is not written and presented well as per the standard of the journal. Further, the work in the manuscript lack scientific novelty as a lot of work for the analysis of methylenecyclopropyl-glycine (MCPG) and pesticides have already been published in last few years. I, therefore, wouldn’t like to recommend this manuscript for the publication PLOS ONE in its present form.

Some comments to improve the quality of the manuscript are as follows:

• The English language and grammar are very poor throughout the manuscript.

• There are a lot of sentences in the introduction which are not properly cited.

• The content of the introduction section should be improved focusing more on MCPG and pesticide toxicity.

• The author should increase the sample size of the urine and plasma samples for this study which can be statistically significant.

• Section 2.6, 2.8 and 2.9 can be merged.

• The urine and plasma samples were not subjected for protein precipitation prior extraction. Any explanation?

• The quality of all the images provided are very poor. please improve.

Reviewer #2: Sinha and colleagues have discussed the mysterious outbreak of Muzaffarpur, Bihar in a sophisticated and chronological order. The paper has been well written and well organized. The introduction and background are reasonable and setting up premises for the paper. This paper concluded that MCPG (methylene cyclopropyl-glycine) and organophosphorus pesticides were the probable culprit of the encephalitis related death in children of Muzaffarpur. While the observations are interesting and potentially useful, I find the paper needs some minor revisions to make it more appealing to the general audience.

1-There are numerous grammatical and typo errors, and the explanations are not clear in some places. For example, in the first paragraph of the introduction “In India, a total of 68.49 metric tons were produced, and 25.40 metric tons were imported in 2012-2013” this sentence is for litchi production or pesticide consumption it is not clear.

2-In the second paragraph of the introduction the author has mentioned the first outbreak year as 2012 instead of 2011 which is not matching what has mentioned in the abstract.

3-In the third paragraph of the introduction, the first line of the paragraph “plant extracte” should be replaced with “plant extract”.

4-In the fourth paragraph of the introduction the full name of KDKM medical clinic and SKMCH hospital should be mentioned.

5-In the fifth paragraph, the seventh line “The sex proportion was seen as 1.2:1 male to female”. It is not clear for which place this data has been mentioned. In the 8th line, reference 14 is not in a proper format.

6-In the material method section, Acetyl-cholinesterase estimation, the author should mention the full name of SLK diagnostics

7-In the material method section, under the subheading “Mass spectrometer instrument control by the analyst program” the last line “Precursor ions……………. confirmation” is not clear and hard to understand.

8-In the result section, in the second paragraph “ac-(methyl cyclopropyl) glycine” should be replaced with “α-methyl cyclopropyl glycines”.

9-In the result section, under the subheading “Isolation of MCPG”, the author should clearly define the respective Rf values for each sample.

10-In the discussion section, in the first paragraph and second-line “ Analysis……………..critically” should be rewritten.

11-In the fourth paragraph of the discussion “toxics” should be replaced with “toxins”.

Reviewer #3: Sinha et al has reported the work titled “A recurring outbreak of litchi fruit consumption among children of Muzaffarpur, Bihar”. Authors have tried to identify toxic compounds in fruit litchi and in urine samples obtained from children. Authors found MCPG in litchi samples and identified DEP in urine samples. In addition to these, they performed acetylcholinesterase biochemical assays. The present report is important study as litchi outbreak occurs every year in Bihar due to its consumption.

1. MCPG is a well reported metabolite in litchi. I was just wondering if authors found any other toxic compounds in the litchi fruit.

2. Did authors find any OP pesticides in litchi fruit? If yes what are they?

3. I would suggest quantifying the MCPG in the litchi samples and quantify DEP in urine samples.

6. PLOS authors have the option to publish the peer review history of their article (what does this mean?). If published, this will include your full peer review and any attached files.

Reviewer #1: No

Reviewer #2: **Yes: **Nagendra Rai

Reviewer #3: **Yes: **Ashish Dwivedi, Scientist, CSIR-IITR, Lucknow

---

## [Author Response · Author response to Decision Letter 0]

28 Oct 2020

Reviewer Comments

Reviewer #1: 

Reviewer Comment 1:

The English language and grammar are very poor throughout the manuscript.

Reply: The whole manuscript was checked for English language and grammar errors and revised accordingly and also the revised manuscript was proofread by a native English speaker and corrected by Grammarly software finally. 

Reviewer Comment 2:

There are a lot of sentences in the introduction which are not properly cited.

Reply: We thank the reviewer for the identification of errors and the same has been rectified in the revised manuscript.

Reviewer Comment 3:

The content of the introduction section should be improved focusing more on MCPG and pesticide toxicity.

Reply: Changes and additions with regards to MCPG and pesticide toxicity have been revised into the manuscript. 

Reviewer Comment 4: 

The author should increase the sample size of the urine and plasma samples for this study which can be statistically significant.

Reply: The samples were collected in the year 2014 during the outbreak as a part of the present investigative study, which has shown interesting findings and further research with large sample size is required. 

Reviewer Comment 5: 

Section 2.6, 2.8 and 2.9 can be merged.

Reply: 

Section 2.6, 2.8 and 2.9 are merged as per the reviewer suggestions (Page NO: 8-9)

Reviewer Comment 6:

The urine and plasma samples were not subjected for protein precipitation prior extraction. Any explanation?

Reply: The method employed in our study was a reported method developed by the CDC Atlanta for the analysis of pesticide metabolites in urine samples.

Reviewer Comment 7: 

The quality of all the images provided are very poor. please improve.

Reply: Image quality has been improved and new images with high resolution are included in the revised manuscript.

Reviewer #2: 

Reviewer Comment 1

There are numerous grammatical and typo errors, and the explanations are not clear in some places. For example, in the first paragraph of the introduction “In India, a total of 68.49 metric tons were produced, and 25.40 metric tons were imported in 2012-2013” this sentence is for litchi production or pesticide consumption it is not clear.

Reply: The details mentioning about production and export was with regard to litchi fruit. The same is now clearly mentioned in the revised manuscript. Also, as per the reviewer's suggestions, manuscript was checked for any ambiguous sentences and revised.

Reviewer Comment 2

In the second paragraph of the introduction, the author has mentioned the first outbreak year as 2012 instead of 2011 which is not matching what has mentioned in the abstract.

Reply: The typographic error of 2012 was corrected to 2011 which is the first outbreak occurred in Bihar. (Page -3, Paragraph 2)

Reviewer Comment 3

In the third paragraph of the introduction, the first line of the paragraph “plant extracte” should be replaced with “plant extract”.

Reply: Typographical error of plant extracte to plant extract has been corrected as per the reviewer suggestion. (Page -3, Paragraph 3)

Reviewer Comment 4

In the fourth paragraph of the introduction the full name of KDKM medical clinic and SKMCH hospital should be mentioned.

Reply: The full name of KDKM (Krishnadevi Deviprasad Kejriwal Maternity Hospital) and SKMCH (Shri krishna Medical College & Hospital) are provided in the revised manuscript. (Page 4, Paragraph 1)

Reviewer Comment 5

In the fifth paragraph, the seventh line “The sex proportion was seen as 1.2:1 male to female”. It is not clear for which place this data has been mentioned. In the 8th line, reference 14 is not in a proper format.

Reply: The necessary modified indicating the proportions and referencing in proper format were made in the revised manuscript as per the reviewer suggestions. (Page 4, Paragraph 1)

Reviewer Comment 6

In the material method section, Acetyl-cholinesterase estimation, the author should mention the full name of SLK diagnostics

Reply: The name SLK diagnostics mentioned in the manuscript represents the name and the diagnostic laboratory has identified no abbreviated full form. However, the estimation of ache is depleted in the manuscript since no significant results were obtained regarding the parameter. 

Reviewer Comment 7

In the material method section, under the subheading “Mass spectrometer instrument control by the analyst program” the last line “Precursor ions……………. confirmation” is not clear and hard to understand.

Reply: This section is a method that has previously developed by us for the estimation of OP metabolites on LCMS/MS in which precursor ions play a major role in the confirmation of mass of compounds to be identified. Further, the same method was applied here for identification of OP metabolites in the urine samples of children and the reference has been cited in the manuscript. (Page -9, Paragraph – 1)

Reviewer Comment 8

In the result section, in the second paragraph “ac-(methyl cyclopropyl) glycine” should be replaced with “α-methyl cyclopropyl glycines”.

Reply: As per the suggestions of the reviewer, α-methyl cyclopropyl glycines has been replaced for ac-(methyl cyclopropyl) glycine in the revised manuscript. (Page 10, Paragraph 3)

Reviewer Comment 9

In the result section, under the subheading “Isolation of MCPG”, the author should clearly define the respective Rf values for each sample.

Reply: The Rf values for each sample was incorporated and rewritten as per the reviewer suggestions in the revised manuscript. (Page 11, Paragraph 1)

Reviewer Comment 10

In the discussion section, in the first paragraph and second-line “Analysis……………. critically” should be rewritten.

Reply: As per the reviewer suggestions the lines were rewritten as, “The litchi fruit samples and biological (urine) samples of the children from Muzaffarpur were collected and analyzed in the present study to critically investigate the cause of the outbreak” in the revised manuscript. (Page 12, Paragraph 2)

Reviewer Comment 11

In the fourth paragraph of the discussion “toxics” should be replaced with “toxins”.

Reply: The word toxics has been replaced with toxins as per the review suggestions in the revised manuscript. (Page 12, Paragraph 3)

Reviewer #3: 

Reviewer Comment 1:

MCPG is a well reported metabolite in litchi. I was just wondering if authors found any other toxic compounds in the litchi fruit.

Reply: We were focusing only on MCPG and pesticide metabolites in the present study and no other toxic compounds were identified. However, it would be interesting and innovative to look beyond and identify if there could be any other compounds that are toxic in litchi fruit.

Reviewer Comment 2:

Did authors find any OP pesticides in litchi fruit? If yes what are they?

Reply: We have extracted the litchi fruits to identify any pesticides but we could not find any significant levels of OP pesticides in any of the fruit samples. 

Reviewer Comment 3:

I would suggest quantifying the MCPG in the litchi samples and quantify DEP in urine samples.

Reply: We thank the reviewer for their valuable comments and suggestions and the DEO metabolites in urine samples were quantified and the levels were found to be 3327 ɳg/mL which was identified in one urine sample. We could not quantify the level of MCPG as we could not get access to the standard of MCPG. (Page 12, Paragraph 1)

---

## [Decision Letter · Decision Letter 1]

23 Nov 2020

PONE-D-20-25356R1

A recurring disease outbreak of Litchi fruit consumption among children of Muzaffarpur, Bihar - A comprehensive investigation on factors of toxicity

PLOS ONE

Dear Dr. Sinha,

Thank you for submitting your manuscript to PLOS ONE. After careful consideration, we feel that it has merit but does not fully meet PLOS ONE’s publication criteria as it currently stands. Therefore, we invite you to submit a revised version of the manuscript that addresses the points raised during the review process.

Please

1) Improve the english of the manuscript 

2) Explain how the present study findings are informative than previously published results

3) Improve the quality of images

We look forward to receiving your revised manuscript.

Kind regards,

Ch Ratnasekhar, Ph.D.

Academic Editor

PLOS ONE

Reviewers' comments:

Reviewer's Responses to Questions

**Comments to the Author**

1. If the authors have adequately addressed your comments raised in a previous round of review and you feel that this manuscript is now acceptable for publication, you may indicate that here to bypass the “Comments to the Author” section, enter your conflict of interest statement in the “Confidential to Editor” section, and submit your "Accept" recommendation.

Reviewer #1: All comments have been addressed

2. Is the manuscript technically sound, and do the data support the conclusions?

Reviewer #1: Partly

3. Has the statistical analysis been performed appropriately and rigorously? 

Reviewer #1: No

4. Have the authors made all data underlying the findings in their manuscript fully available?

Reviewer #1: Yes

5. Is the manuscript presented in an intelligible fashion and written in standard English?

Reviewer #1: Yes

6. Review Comments to the Author

Reviewer #1: The revised manuscript entitled "A recurring disease outbreak of Litchi fruit consumption among children of Muzaffarpur, Bihar - A comprehensive investigation on factors of toxicity" submitted has been improved very much from the earlier submission. The maximum comments raised have been addressed apart from the language is still not satisfactory along with the quality of images submitted to journal is very poor.

7. PLOS authors have the option to publish the peer review history of their article (what does this mean?). If published, this will include your full peer review and any attached files.

Reviewer #1: No

---

## [Author Response · Author response to Decision Letter 1]

9 Dec 2020

Comment 1: Improve the english of the manuscript 

Response: We thank the editor and reviewers for their comments and help us in improving our manuscript. The English and grammar have been corrected using Editage services and the certificate has been attached with cover letter for your kind reference. 

Comment 2: Explain how the present study findings are informative than previously published results

Response: Although similar studies were conducted previously, they focused on identifying MCPG, a known toxin found in litchi. However, the present study considered other sources of toxicity that can occur due to litchi consumption. The use of OP pesticides in Bihar for litchi cultivation called for a thorough investigation. Therefore, litchi fruit samples and urine samples from the children from Muzaffarpur were collected and analyzed in the present study to critically investigate and identify the toxic compounds (MCPG and OP pesticide metabolites), which may have caused the fatalities. The above has also been included in the revised manuscript. 

Comment 3: Improve the quality of images

Response: The original images have been uploaded along with the revised manuscript.

---

## [Editor Report · Decision Letter 2]

17 Dec 2020

A recurring disease outbreak following litchi fruit consumption among children in Muzaffarpur, Bihar - A comprehensive investigation on factors of toxicity

PONE-D-20-25356R2

Dear Dr. Sinha,

We’re pleased to inform you that your manuscript has been judged scientifically suitable for publication and will be formally accepted for publication once it meets all outstanding technical requirements.

Kind regards,

Ch Ratnasekhar, Ph.D.

Academic Editor

PLOS ONE
---

## [Editor Report · Acceptance letter]

21 Dec 2020

PONE-D-20-25356R2 

A recurring disease outbreak following litchi fruit consumption among children in Muzaffarpur, Bihar - A comprehensive investigation on factors of toxicity 

Dear Dr. Sinha:

I'm pleased to inform you that your manuscript has been deemed suitable for publication in PLOS ONE. Congratulations! Your manuscript is now with our production department. 

Kind regards, 

on behalf of

Dr. Ch Ratnasekhar 

Academic Editor

PLOS ONE